# Gut Microbiota Linked with Reduced Fear of Humans in Red Junglefowl Has Implications for Early Domestication

*Lara C. Puetz,\* Tom O. Delmont, Ostaizka Aizpurua, Chunxue Guo, Guojie Zhang, Rebecca Katajamaa, Per Jensen, and M. Thomas P. Gilbert\**

Domestication of animals can lead to profound phenotypic modifications within short evolutionary time periods, and for many species behavioral selection is likely at the forefront of this process. Animal studies have strongly implicated that the gut microbiome plays a major role in host behavior and cognition through the microbiome–gut–brain axis. Consequently, herein, it is hypothesized that host gut microbiota may be one of the earliest phenotypes to change as wild animals were domesticated. Here, the gut microbiome community in two selected lines of red junglefowl that are selected for either high or low fear of humans up to eight generations is examined. Microbiota profiles reveal taxonomic differences in gut bacteria known to produce neuroactive compounds between the two selection lines. Gut–brain module analysis by means of genome-resolved metagenomics identifies enrichment in the microbial synthesis and degradation potential of metabolites associated with fear extinction and reduces anxiety-like behaviors in low fear fowls. In contrast, high fear fowls are enriched in gut–brain modules from the butyrate and glutamate pathways, metabolites associated with fear conditioning. Overall, the results identify differences in the composition and functional potential of the gut microbiota across selection lines that may provide insights into the mechanistic explanations of the domestication process.

## 1. Introduction

Phenotypic plasticity is conventionally viewed as being directly encoded by the genome, either due to conventional genetic, or even epigenetic, variation. However, a growing body of evidence shows that microorganisms residing in the gut of animals (their microbiota) may play a crucial role on their adaptive capacity.[1] The gut microbiota is a major actor in the nutrition, health, physiology, and behavior of complex animals,[2–7] and the close connection between hosts and their gut microbiota has led researchers to assert that their combined activities represent both a shared target for natural selection and a driver of adaptive responses.[8–14] Adapting to novel environments involves modification of biological systems in an effort to maintain fitness in response to a stressing agent. Ample evidence now portrays the gut microbiome as a dynamic biological system that

L. C. Puetz, O. Aizpurua, M. T. P. Gilbert
Center for Evolutionary Hologenomics
GLOBE Institute
University of Copenhagen
Copenhagen 1353, Denmark
E-mail: lara.puetz@science.ku.dk; tgilbert@sund.ku.dk

T. O. Delmont
Génomique Métabolique
Genoscope
Institut François Jacob
CEA
CNRS
Univ Evry
Université Paris-Saclay
Evry 91057, France

C. Guo, G. Zhang
China National GeneBank
BGI-Shenzhen
Shenzhen 518083, China

G. Zhang
Villum Center for Biodiversity Genomics, Section for Ecology and Evolution, Department of Biology
University of Copenhagen
Copenhagen 2100, Denmark

G. Zhang
State Key Laboratory of Genetic Resources and Evolution
Kunming Institute of Zoology
Chinese Academy of Sciences
Kunming 650223, China

G. Zhang
Center for Excellence in Animal Evolution and Genetics
Chinese Academy of Sciences
Kunming 650223, China

R. Katajamaa, P. Jensen
IFM Biology, AVIAN Behaviour Genomics and Physiology Group
Linköping University
Linköping 58330, Sweden

M. T. P. Gilbert
Department of Natural History, NTNU University Museum
Norwegian University of Science and Technology (NTNU)
Trondheim 7491, Norway

responds to environmental changes and is able to adapt to novel conditions.

Domestication is an evolutionary process of adaptation that can lead to profound phenotypic modifications in animals within short time periods. The central selective pressure is thought to be imposed through the new social environment, and the initial exposure to humans is known to induce rapid behavioral change in wild animals during the early stages of the domestication process.[15] A reduction in acute fear and long-term stress toward humans is a shared feature among domesticated animals[16,17] and often a prerequisite to successful breeding in captivity.[18] Contemporary research has strongly implicated the gut microbiome in brain development, host behavior, and cognition, through its ability to produce and modify metabolic, immunological, and neurochemical factors in the gut that can impact host physiology and the nervous system.[19,20] Consequently, we hypothesized that host gut microbiota may be one of the earliest phenotypes to change as wild animals are domesticated.

To test this hypothesis, we examined the association of gut microbiome features with two selected lines of red junglefowl (the ancestors of domestic chickens) that were bred at the University of Linköping (Sweden) solely for either high or low fear of humans, over the course of eight generations. In this study, red junglefowl were reared under identical controlled environments and housed in mixed group pens.[21–23] As such, we were able to uniquely assess whether the community composition of the gastrointestinal microbiota differed between the two behavioral phenotypes without introducing the confounding effects of environment, including diet, on microbial community structure. We additionally performed functional analysis of genome-resolved metagenomes to assess the potential of the microbiome to produce or degrade neuroactive compounds that may act as mediators of microbiota–gut–brain signaling.[24] In this exploratory study, we hypothesize that the gut microbiome profiles and their neuroactive potential are associated with red junglefowl behavioral phenotypes, which may provide novel insights into the mechanistic explanations of the domestication process.

## 2. Results

### 2.1. Low Fear Red Junglefowl Are Consistently Depleted in *Lactobacillales* and Enriched in *Clostridiales*

To facilitate the detection of low abundant taxa and create fine-scale community composition profiles of the gut microbiota of the two red junglefowl behavioral phenotypes, a 16S rRNA gene amplicon survey was first performed on fecal samples representing high (HF) and low (LF) fear individuals selected in the sixth (S6) and seventh (S7) generations (2016 collection year; $n$ = 22) and from the eighth generation (S8) (2018 collection year; $n$ = 29). A total of 2582 amplicon sequence variants (ASVs) were identified, 23 of which were differentially abundant between the high and low fear behavioral phenotypes, and an additional 34 ASVs were discriminant to either selection line in both sampling years (Figure S1 and Table S1, Supporting Information). Although many occurred within the rare biosphere, a noteworthy number of differences occurred in abundant and highly prevalent taxa (Figure S1, Supporting Information). Most of these differences were *Firmicutes* ASVs ($n$ = 42), the most abundant phylum (Figure S1, Supporting Information). Order level differences identified that *Clostridiales* were significantly more abundant in the low fear selection line (**Figure 1**; Wald test, $t$ value = 3.1, $p$ value = 0.003) and further analysis at the ASV level revealed a significant enrichment of ASVs belonging to the *Ruminococcaceae* (genera *Subdoligranulum* and *UCG-014*), *Lachnospiraceae*, *Clostridiaceae_1*, and *Peptostreptococcaceae* families within this order, the latter two of which had ASVs exclusively found in this line (**Figure 2**; Table S1, Supporting Information). Conversely, the *Lactobacillales* order was significantly more abundant in the high fear selection line (Figure 1; Wald test, $t$ value = -3.7, $p$ value = 0.0006), including many ASVs from the *Lactobacillus* genus, but additionally a *Streptococcus* sp. and several unknown ASVs from this order (Figure 2; Table S1, Supporting Information).

Low fear fowls were also enriched in several *Bacteroidetes* ASVs (Figure 2A; Table S1, Supporting Information). Further, many ASVs were discriminant to the LF selection line including ASVs from *Actinobacteria* (*Rathayibacter* and unknown genera), *Gammaproteobacteria* (*Pseudomonas* and unknown genera), *Verrucomicrobia* (*Cerasicoccus* sp.*)*, and finally a Proteobacteria (*Helicobacter* sp.*)* (Figure 2B). Conversely, high fear fowls were enriched in *Synergistes* and *Sutterella* ASVs, as well as several unknown ASVs from the *Erysipelotrichales* order (Figure 2).

### 2.2. Genome-Resolved Analysis of Red Junglefowl Fecal Samples Yielded 194 Manually Curated Microbial Genomes

Shotgun metagenomic sequencing of the 51 fecal samples yielded 6.1 billion high-quality short-reads, of which 19–91% mapped to the chicken genome (Table S2a, Supporting Information). We then performed a comprehensive genome-resolved metagenomic survey of the microbial populations through coassembly of these data after host-derived reads were removed. Subsequent manual binning within the anvi'o[25] framework using differential coverage across all samples resulted in 194 nonredundant metagenome-assembled genomes (MAGs) (**Figure 3**A; Figure S3 and Table S2b, Supporting Information). Six samples, all males, had a sequencing depth of less than 10 million single-end reads (<1 Gb) after quality control and were removed from downstream analyses (as per recent recommendations[26]) due to the large amount of host DNA (>70%) minimizing microbial signal. However, for this study, we note that each behavioral phenotype was equally represented by the MAGs (HF: $n$ = 22, LF: $n$ = 23) (Figure S4, Supporting Information). The genomic database revealed that two MAGs were affiliated to Archaea, while all remaining MAGs ($n$ = 192) were affiliated to known bacterial orders, including 40% identified at the species level (average nucleotide identity >95%). We identified 15 bacteria phyla; the most frequent MAGs being Firmicutes ($n$ = 94 MAGs), followed by Bacteroidetes ($n$ = 57), Actinobacteriota ($n$ = 14), Proteobacteria ($n$ = 13), Patescibacteria ($n$ = 6), Tenericutes ($n$ = 4), Cyanobacteria ($n$ = 1), Fusobacteria ($n$ = 1), Deferribacteres ($n$ = 1), Spirochaetes ($n$ = 1), Synergistetes ($n$ = 1) and Verrucomicrobia ($n$ = 1) (Figure 3A). Complete taxonomic assignments using the Genome Taxonomy Database Toolkit (GTDB-Tk)[27] are found in Table S2b (Supporting Information).

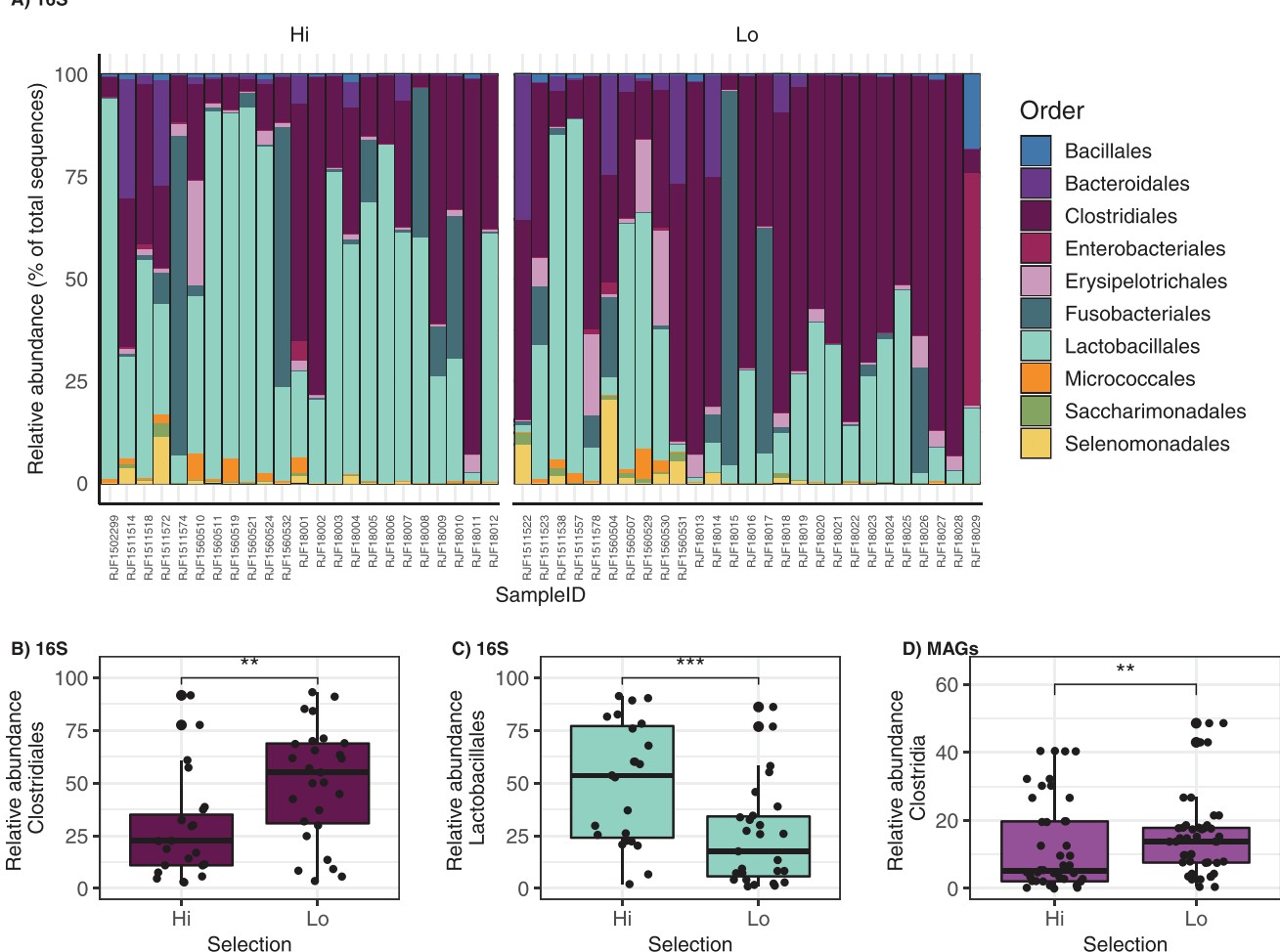

**Figure 1.** Relative abundance of bacteria detected in fecal microbiota of red junglefowl selected for high fear and low fear towards humans. Relative abundance of order-level bacterial in 16S rRNA amplicon sequence variants detected in 23 high fear and 27 low fear red junglefowl fecal samples: A) Top 10 most abundant bacteria orders and differentially abundant. B) *Clostridiales* (Wald test, t value = 3.1, p value = 0.003) and C) *Lactobacillales* (Wald test, t value = −3.7, p value = 0.0006) orders between the high and low fear selection lines. D) Differential abundance of the class *Clostridia* (Wald test, t value = 3.3, p value = 0.002) observed in the metagenome-assembled-genomes (MAGs) detected in 22 high fear and 23 low fear red junglefowl fecal samples. Microbial community composition was modeled at the order and class level for B,C) the 16S and D) shotgun sequence data respectively by fitting the beta-binomial regression model implemented in the "corncob" package in R. Differentially abundant taxa were considered significant using the parametric Wald test with a controlled false discovery rate (p value cutoff <0.05) ** p ≤ 0.01, *** p ≤ 0.001.

We subsequently modeled microbial abundance at the class level to reveal a significant enrichment of *Clostridia* MAGs in low fear individuals (Wald test, t value = 3.3, p value = 0.002), corroborating similar findings to the 16S amplicon data presented above (Figure 1D). Further, two MAGs were discriminant to the low fear selection line in both collection years and affiliated to *Helicobacter magdeburgensis* (similarly identified in the 16S sequence data; Figure 2B) and an unknown *Limosilactobacillus* population within the *Firmicutes* (Figure 3). Conversely, *Limosilactobacillus ingluviei* was significantly enriched in the high fear fowls, and a *Clostridiales* MAG with no culture representatives (*UBA9475* sp. 003534045) were exclusively found in this line across both years (Figure 3). All four MAGs were less than 2% redundant and near complete (>92%) (Figure 3A).

## 2.3. Gut–Brain Modules Well-Represented in the Gut Microbiota of the Red Junglefowl

In order to describe the neuroactive potential of gut microbiota in relation to gut–brain interactions in the red junglefowl, we applied a module-based framework previously developed in human microbiome research.[24] This framework identifies microbial pathways that metabolize molecules that have the potential to interact with the host nervous system. We found 41 out of the 56 annotated gut–brain modules (GBMs) known to produce or degrade neuroactive compounds and these were spread widely across the phylogenic range of the red junglefowl MAGs (Figure 3A). Several phylogenetic GBM hotspots (high prevalence of GBMs) were observed in the *Proteobacteria* (all

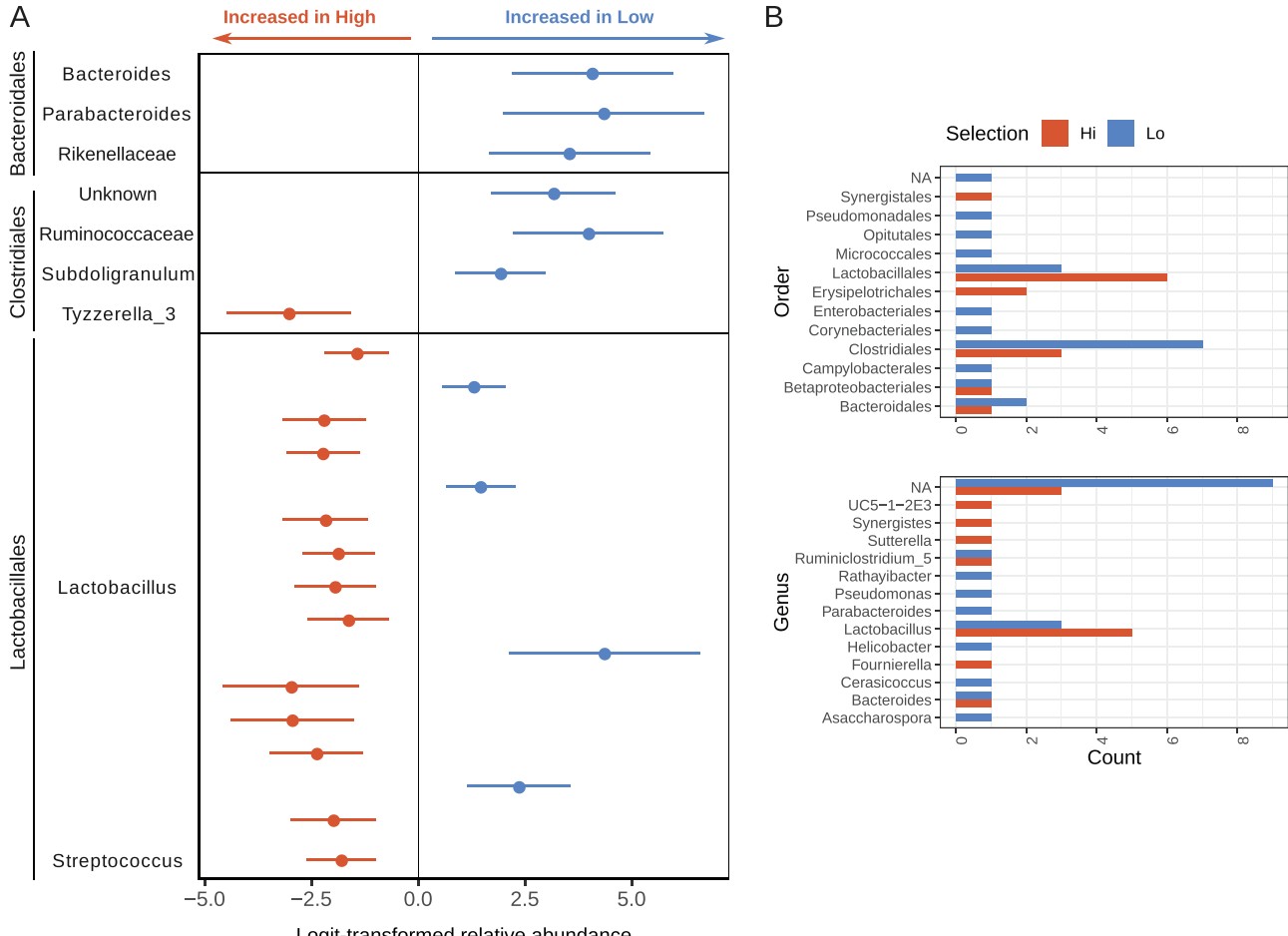

**Figure 2.** Differential abundance of bacterial 16S ASVs between behavioral phenotypic groups of red junglefowl. A) Differences in the abundance of ASVs detected in 50 red junglefowl fecal samples, grouped by order-level taxonomic classification, between high ($n = 23$) and low fear ($n = 27$) selection lines estimated with corncob using the Wald test with a controlled false discovery rate. ASVs enriched in either selection line were categorized by family-level classification and annotated on the right y-axis. B) Discriminant number ASVs, exclusively found in either selection line, grouped by order and genus level classifications.

*Gammaproteobacteria*) and *Actinobacteria* phyla, whereas MAGs affiliated to the *Patescibacteria* phylum, known for its streamlined genomes lacking several metabolic pathways, carried no more than one, if any, of the GBMs (Figure 3A). None of the GBMs were ubiquitous (present in >90% of the MAGs), which can in part be explained by the incomplete nature of MAGs, and nearly half ($n = 20$) were rare (prevalence < 5% of the MAGs) (**Figure 4**). Notably, propionate II synthesis was present in 22% of the red junglefowl MAGs ($n = 194$) yet rarely found in human reference genomes used to originally validate this framework ($n = 532$).[24] Additionally, we identified two modules, vitamin K synthesis II and acetate synthesis III, that were detected in fowl MAGs but absent in humans. Serotonin, dopamine, acetylcholine, polyunsaturated fatty acids (PUFAs), and propionate I synthesis were absent in RJF MAGs in addition to glutamate I degradation. Most GBMs ($n = 32$; 78%) were present in over 50% of all RJF samples and two were rare, namely kynurenine synthesis (2%) and acetate synthesis II (8%) (Figure 4C).

### 2.4. Behavioral Selection Induces Shifts in the Functional Potential of the Red Junglefowl Gut Microbiome

To explore whether neuroactive compound metabolism could be associated with behavioral selection, we assessed the detection of GBMs in MAGs that were significantly enriched or discriminant to a behavioral phenotypic group. Based on the GBM framework applied to all RJF samples, there were no significant differences in the overall differential abundance of GBMs (Figure 4C); however, when applied to each of the four significantly enriched or discriminant MAGs, we identified nine associations with either of the selection lines, three of which were GBMs rarely found in the red junglefowl MAGs (present in <5% of all MAGs) (Figure 3B). Two of the three rare GBM associations were short-chain fatty acids (SCFA), metabolites produced by gut bacteria, and suspected of playing key roles in the microbiota–gut–brain axis. Interestingly, high fear fowls were enriched in GBMs from the butyrate pathway, namely butyrate synthesis I and II, the latter of

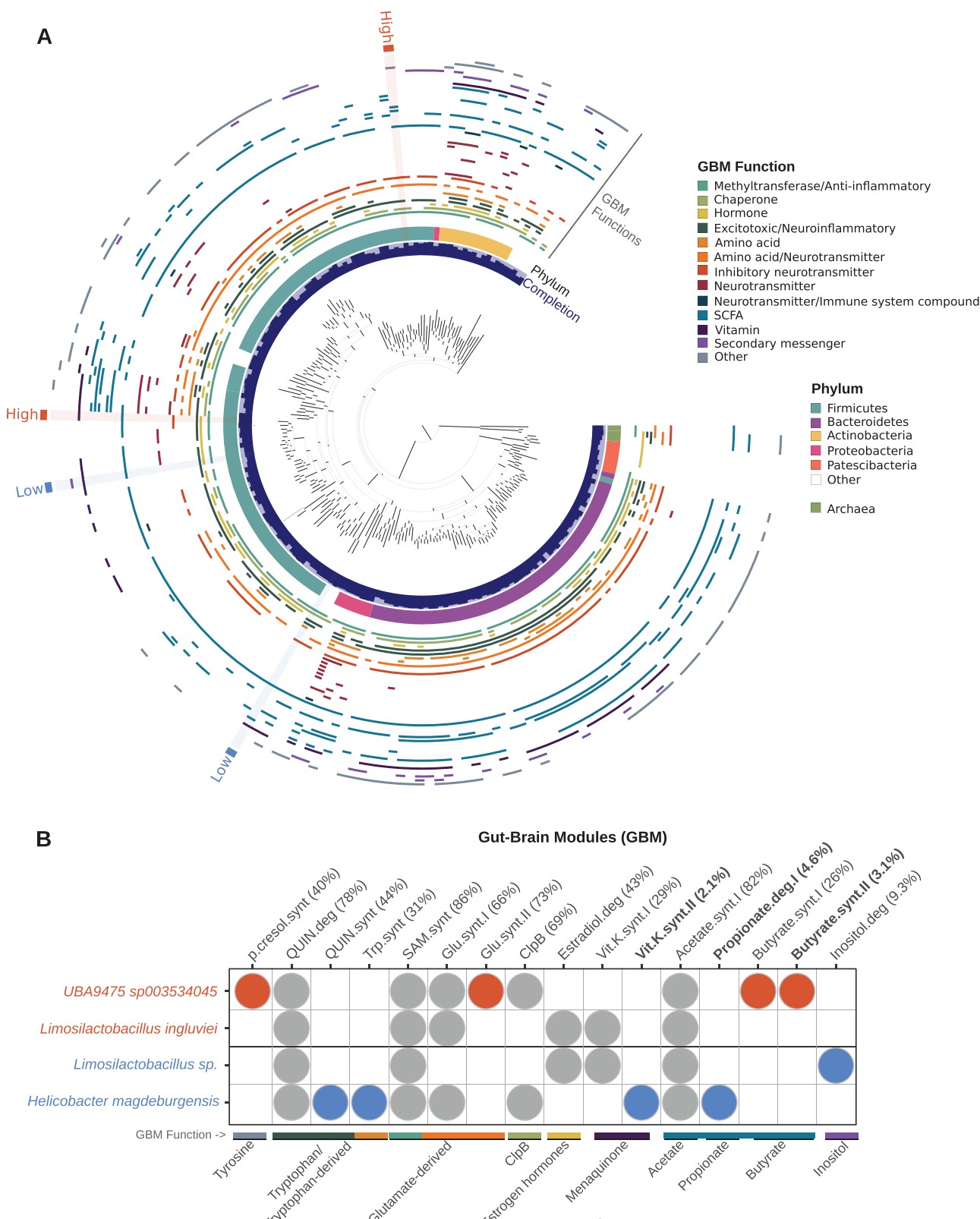

**Figure 3.** Gut–brain module (GBM) distribution in red junglefowl metagenome-assembled genomes (MAGs). A) Maximum-likelihood phylogenetic tree comprising 194 gut-associated MAGs identified in 45 red junglefowl fecal samples (HF: *n* = 22, LF: *n* = 23). The innermost circular layer represents the percent completion of each genome followed by the associated phylum in the second layer. The following middle layers represent the 41 gut–brain

which was found in only 3.1% of the MAGs, whereas the low fear selection line was enriched in the rare SCFA module from the propionate pathway, propionate degradation I (Figure 3B). Four GBMs were additionally enriched in the low fear selection line including two from the tryptophan/tryptophan derived pathways, namely quinolinic acid and tryptophan synthesis, inositol degradation, and finally the rare GBM menaquinone synthesis (vitamin K) II. Conversely, p-Cresol synthesis from the tyrosine pathway and glutamate synthesis II from the glutamate-derived pathway were GBMs associated with the high fear selection line (Figure 3B).

## 3. Discussion

Here we demonstrate that despite having been subjected to relatively few generations of bidirectional selection for low or high fear toward humans, significant differences exist in the community composition of the red jungle fowl gut microbiota. Recent studies have identified that ecological shifts during domestication, specifically changes in diet, can have major impacts on the gut microbiota.[28–31] Although diet undoubtedly plays a significant role in shaping the gut microbiome, the controlled environments used in this study allow us to rule out diet as the driver of phenotypic differences between selection lines during the early stages of domestication in red junglefowl. We further posit that differences in microbial community composition may reflect altered interactions within the microbiota–gut–brain axis.

During early domestication, one central selective pressure is thought to be the new social environment (human encounters), in which specific animals exhibiting excessive fear and stress reactions toward humans would have been unlikely to thrive, thus reproduce. Experimental studies using germ-free mice have demonstrated that the presence of a functional gut microbiota affects synaptic plasticity[32] as well as fear memory retention[33] and extinction in the host.[34] Further, shifts in community composition of certain gut microbiota, namely an increase in *Lactobacillus* and a decrease in *Clostridium*, have been identified to be highly correlated with host learning and memory in rats.[35] As such, these observations would suggest that gut microbiota may have been capable of intensifying or reducing fear-like behaviors and memory retention during the domestication of at least some wild animals. As the potential bacterial species that could be responsible for these phenotypes, as well as the mechanisms underlying their presentation, remain elusive, it is interesting that we found significant order-level increases of *Clostridiales* and *Bacteroidales* in the gut microbiota of red junglefowl in low fear animals and significant enrichment of *Lactobacillales* in high fear animals.

Increased prevalence of *Lactobacillus* in the gut microbiota of our high fear selection animals is consistent with observations in phobic dogs undergoing intense states of fear,[36] supporting a potential role for this genus in red junglefowl fear phenotypes. Furthermore, this bacterial genus has been repeatedly implicated in enhanced fear memory retention in mammals.[37–39] *Lactobacillus rhamnosus JB-1*, for example, has been associated with both aggressor avoidance behaviors as well as enhanced fear memory recall in mice.[37,38] These phenotypic changes in mice were accompanied by gene expression profiles in the brain related to fear memory, impaired fear extinction, and stress responses.[37,38]

In contrast to our observed association of *Lactobacillus* with fear behavior, we observed that *Clostridiales* were enriched in the low fear fowl and several unknown *Clostridiales* species have been associated with reduced stress in Japanese quail.[40] This observation is interesting in light of previous findings associating these taxa with impaired fear memory in obese mice, including members of the orders *Lachnospiraceae* and *Ruminococcaceae*.[41] *Clostridiales* are also known to promote host serotonin biosynthesis[42] and higher blood serotonin has been observed in the low fear red junglefowl males.[43] Although serotonin cannot cross the blood-brain barrier, peripheral serotonin is important in neurodevelopment.[44,45] The gut microbiota has been found to play a key role during neonatal neurodevelopment in mice in which microbiota-derived signals induce learning-related plasticity, including fear extinction learning, that persists into adulthood.[34] While Chu and co-workers[34] inferred that a more diverse gut microbiota was required for normal fear extinction behavior, it is possible that certain individuals within the community, such as the *Clostridiales* enriched in the low fear fowls here, may play a more significant role in the fear extinction process. We additionally observed an increased abundance of *Bacteroides* and *Parabacteroides* in low fear fowls, both of which are γ- aminobutyric acid (GABA) producing bacteria.[46] GABA is an inhibitory neurotransmitter that can cross the blood-brain barrier,[47–49] although to what extent remains debatable, and GABA signaling has been implicated in fear extinction learning.[50,51] Notably, increased abundance in *Bacteroides* and *Parabacteroides* has not only been associated with decreased levels of anxiety, distress, and irritability in healthy women but also increased gray matter in the cerebellum,[52] an associated phenotypic trait consistently observed in low fear red junglefowl.[53] Together these data suggest that gut microbiota significantly enriched in low fear fowls may have the capacity to impair fear memory, promote fear extinction, or prevent anxiety-like behaviors, all of which would be relevant to overcoming a fear response toward humans during domestication.

To explore variation in neuroactive metabolic potential resulting from differences in gut microbiota between the two behavioral selection lines, we applied a module-based analytical framework to our metagenomic data.[24] For the high fear junglefowl, our analysis of gut–brain modules revealed the increased functional potential of butyrate synthesis I and II, p-Cresol synthesis, and glutamate synthesis, suggesting a role for these

modules (GBMs) detected in the collections of MAGs and categorized by functional association. Six MAGs were differentially abundant or exclusive to a behavioral phenotypic group and highlighted on the tree based on the enrichment in either the high fear (orange) or the low fear (blue) selection line. B) Detection of GBMs in the six differentially occurring MAGs highlighted in (A). The numbers in parentheses correspond to the frequency of the GBM found in all MAGs and values in bold were considered rare (present in <5% of the MAGs). Pathways and functions were annotated for each GBM along the bottom x-axis. GBMs exclusively present in MAGs enriched in the high fear group are highlighted in orange and those in the low fear group are highlighted in blue.

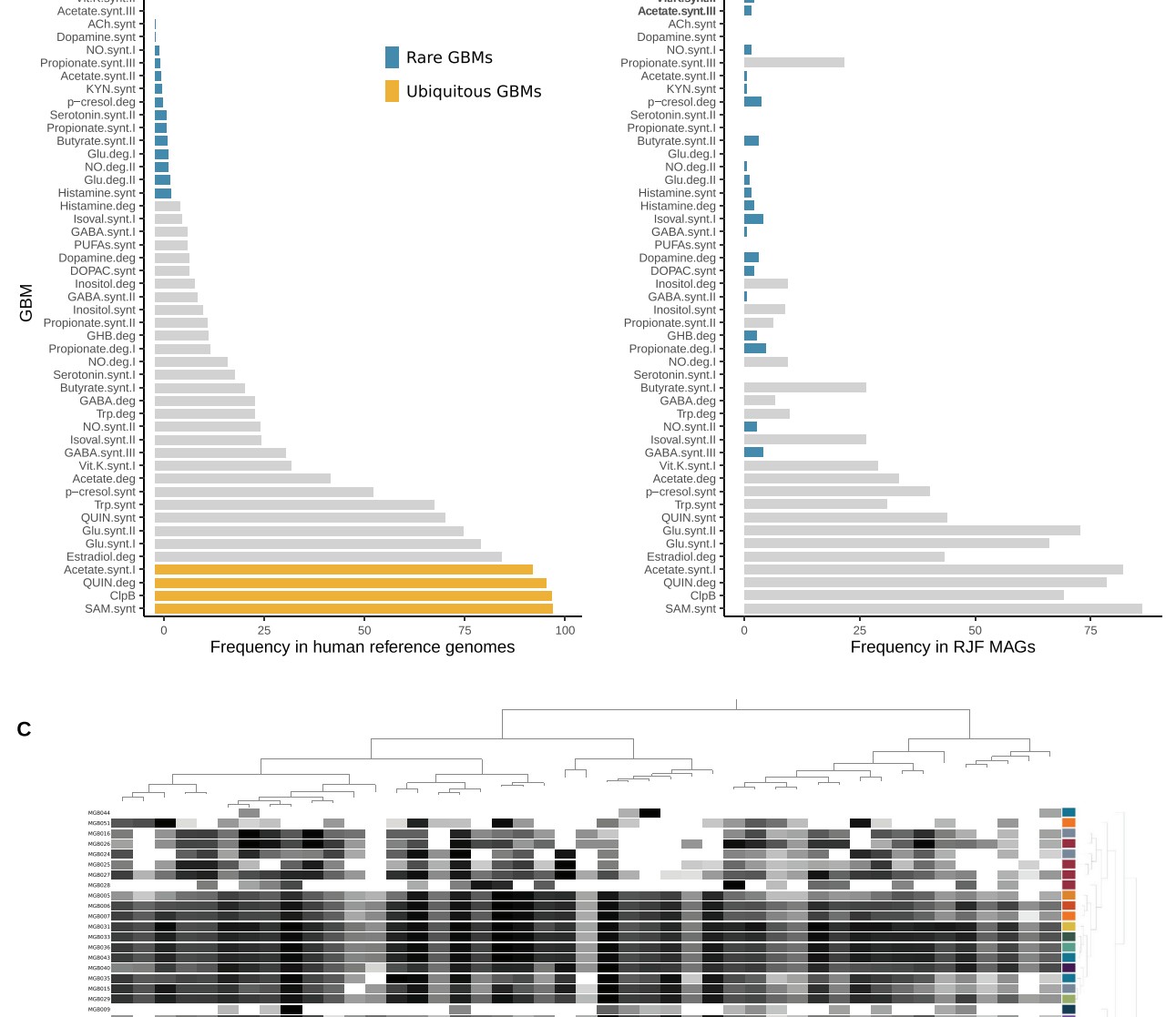

**Figure 4.** Comparison of gut–brain module (GBM) distribution in human and red junglefowl microbial genomes. GBM detection frequency in A) human gut-associated microbial reference genomes (*n* = 532) drawn from Valles-Colomer et al.[24] and previously used to validate the GBM framework and B) red junglefowl MAGs (*n* = 194) identified from fecal samples in the current study. Rare (<5% of genomes) and ubiquitous GBMs (>90% genomes) are highlighted, while others are in gray. GBMs found exclusively in red junglefowl MAGs are in bold. Figure modified from Valles-Colomer et al.[24] C) Heatmap of the $\log_{10}$ frequency of 41 GBMs detected in 45 fecal metagenomes of red junglefowl selected for high and low fear toward humans. Clustering of GBMs and metagenomes is based on GBM frequency (Euclidean distance and Ward linkage) and the data were visualized using anvi'o.

microbial-induced pathways in fear phenotypes. Butyrate can cross the blood-brain barrier[54] and has been implicated in both neuronal plasticity and long-term fear memory formation via increased histone acetylation in the brain.[55,56] Further, increased levels of both p-Cresol and glutamate have been observed in the gut, blood, urine, and feces of children with autism.[57,58] The microbial metabolite p-Cresol specifically has also been shown to induce autistic-like behaviors in at least various mammal species (similar data are lacking for birds), including enhanced fear and anxiety-like behaviors.[59–62] Last, glutamate is the main excitatory neurotransmitter in the brain, and glutamatergic signaling plays an important role in fear conditioning in at least mammals[63,64] (as above, similar data are lacking for birds). Although glutamate does not enter the brain across the blood-brain barrier,[65] the level of glutamate in the blood is positively correlated with that in the brain,[66] suggesting that gut microbiota may play a role in modulating glutamatergic signaling in the brain. With regard to the above (and indeed several subsequent) points, we note that although many of the examples we provide here are from mammalian studies, as we are unaware that similar data exist from avian studies, the deep evolutionary conservation of the gut–brain axis in vertebrate phylogeny[67] and the conserved properties of some chemical signaling, such as $GABA_A$ receptor subunits among vertebrates,[68] suggest the potential to translate these findings to the red junglefowl.

For the low fear junglefowl line, our gut–brain module analysis identified five enriched neuroactive compounds, including vitamin K2 synthesis and inositol and propionate degradation. Circulating levels of propionate and vitamin K have been associated with fear extinction and reduced anxiety-like behaviors in rodents, respectively.[34,69] Furthermore, dietary inositol decreased fearfulness in chickens[70] and levels of inositol in the brain have been associated with synaptic plasticity and episodic memory.[71,72] Although the mechanism by which the gut microbiota influences the brain via the production and degradation of these neuroactive compounds remains elusive, it is possible that they have neuromodulatory effects relevant to behavioral shifts observed during domestication.

We additionally found several associations of the gut microbial community composition with other correlated phenotypes previously documented in the two red junglefowl selection lines. When selecting solely on fearfulness toward humans, red junglefowl displays a range of associated phenotypic changes including increased feeding efficiency, growth, and reproductive output in low fear relative to high fear animals.[21,22,43] Similar shifts in microbial composition to those reported in this study have been documented between chickens with high and low feed efficiency; namely, higher feeding efficiency associated with *Clostridiales* and propionate producing bacteria[73–75] and lower feeding efficiency associated with *Lactobacillus*,[74–76] consistent with low fear and high fear junglefowl phenotypes, respectively. Additionally, the potential synthesis of tryptophan was associated with low fear fowls in the GBM analysis. Although the role of bacterially derived tryptophan in host physiology remains unknown, dietary tryptophan is known to improve reproductive capacity and growth performance in livestock[77] in addition to facilitating a reduced stress response in animals,[78] consistent with what we see in the low fear fowls. Together these data suggest that community-level shifts in gut microbiota during domestica-

tion may alter the physiology of the host through their contribution to metabolism and nutrient absorption, even without dietary changes.[4,5]

Our results identified differences in the composition and functional potential of the gut microbiota across selection lines that might provide insights into the mechanistic explanations of the domestication process. While we do not attempt to address a causative role in the observed relationships, these findings provide the foundation for future studies. We also note that the data generated here do not allow us to differentiate whether the gut microbiota is under selection themselves or that they are responding to selection on host traits, namely behavior. Stress and anxiety-like behaviors, for example, can induce change in the gut microbiome[79–84] possibly through several mechanisms that alter the physicochemical properties of the intestinal habitat.[85] Furthermore, a small, though a potentially not insignificant portion of the gut microbiome is almost certainly determined by host genetics,[86] and it remains unresolved whether host genetic factors also play a role in shaping the gut microbiome during the domestication process, although we hypothesize that it may well be plausible. Likely the behavioral phenotype selected on during domestication reflects a convergence of microbial, host, and environmental factors. However, we hope our study can be used to lay the groundwork for future experimental work, including other layers of data such as host and microbiome transcriptome and metabolome information, as well as fecal transplant experiments, to describe the role of the gut microbiome in the domestication process and more broadly in ecoevolutionary dynamics.

## 4. Experimental Section

*Animals and Sample Collection*: All experimental protocols performed on the red junglefowl were approved by Linköping Council for Ethical Licensing of Animal Experiments, ethical permit to Per Jensen, number 14916-2018 (Linköping, Sweden). Animal handling experiments were conducted in accordance with the approved guidelines. Gut microbiota were sampled from adult red junglefowl (*Gallus gallus*) that were bidirectionally selected over eight generations based solely on their fear response toward humans in a long-term study at the Linköping University, Sweden. For a detailed description of the breeding scheme and selection see refs. [21,22], and the housing conditions of animals specific to the three generations used in this study see ref. [23]. Briefly, the chickens originated from two unrelated zoo populations, Copenhagen Zoo (Denmark) and Götala Research Station (Sweden), and were interbred for two generations to create the initial outbred parental (P0) generation. The parental generation was then used to select for the most fearful (high fear; HF) and least fearful (low fear; LF) individuals toward humans in subsequent generations using a standardized fear of human test when the birds were 12 weeks old. Red junglefowl from both high and low fear selection lines were hatched and reared together under standardized conditions in mixed groups in the same pens for a given generation and received food and water ad libitum. The commercially available conventional chicken feed for red junglefowl remained consistent among years.

Fecal samples were collected and preserved in RNAlater stabilization solution from HF and LF selection lines in the sixth (S6) and seventh (S7) generations in the fall of 2016 (HF: $n = 11$, LF: $n = 10$) and from the eighth generation (S8) in the winter of 2018 (HF: $n = 12$, LF: $n = 17$).

*DNA Extraction*: Prior to DNA extraction, RNAlater was removed with centrifugation (13 000 g for 10 min) and the pellet was washed twice with 1 mL of PBS. DNA was extracted from approximately 150 mg of fecal sample using the DNeasy PowerSoil Kit DNA (Qiagen, Venlo, NL) following the manufacturer's protocol with several modifications. Samples were incubated for 10 min at 65 °C after adding Solution C1 and bead beaten

for 10 min at 30 Hz using a TissueLyser II (Qiagen, Hilden, Germany). Purified DNA was incubated in Solution C6 for 15 min at 37 °C before the final elution spin. Four negative controls (i.e., all reagents except sample continued in the workflow from extraction to sequencing as any other extracts) were included in order to check for potential reagent contamination

*Bacterial Community Composition from 16S rRNA Amplicon Sequencing*: A dual indexed PCR approach was used to target the V3-V4 variable region of the bacterial 16S rRNA gene (≈465 bp) for all fecal samples (*n* = 50) using the primer pair Bact-341F (5′-CCTAYGGG RBGCASCAG-3′) and Bact-806R (5′-GGACTACNNGGGTATCTAAT-3′) with Illumina Nextera overhang adapters (Illumina Inc., San Diego, CA, USA).[87–90] PCR was performed in triplicates and pooled prior to indexing PCR for each individual in order to reduce PCR bias. Pooled libraries were sequenced on an Illumina MiSeq platform using 250PE. Full methodological details can be found in Supporting Information.

Illumina adapter and primer sequences were removed from the 16S metabarcoding sequence data using cutadapt v.2.6[91] and subsequently analyzed using the program DADA2 v.1.12.1[92] and R v.3.6.1[93] to infer amplicon sequence variants (ASVs). Complete code was modified from ref. [94]. Briefly, forward and reverse reads were trimmed to 230 bp. The entire dataset was used to define an error rate at each base pair, and all sequences were denoised using the pooled approach to increase the likelihood of resolving rare sequence variants. Forward and reverse reads were merged, and any pair without perfect overlap and <400 bp was removed prior to chimeric sequence filtering. Each ASV was annotated with the RDP Bayesian classifier[95] against the SILVA database[96] to produce a 16S amplicon taxa table. All subsequent analyses were done in R v.3.6.3 unless otherwise stated.[97] ASV data were preprocessed with the phyloseq package v.1.30.0,[98] and potential contaminants were assessed with the decontam package v 1.6.0.[99] Only samples with >10000 reads and ASVs with more than three observations were included in downstream 16S data analysis.

*Metagenomic Shotgun Sequencing*: Shotgun metagenome data were prepared with genomic DNA using various methods. Libraries were built on six samples using a NEBNext protocol, during which metagenomic DNA was fragmented to an average length of ≈350 bp using the Bioruptor XL (Diagenode, Inc.), with the profile of eight cycles of 15 s of sonication and 90 s of rest. Sheared DNA was converted to BGIseq sequencing technology compatible libraries using NEBNext library kit E6070L (New England Biolabs) and blunt-ended BGISEQ-500-compatible adapters AD1 and AD2.[100] For all other samples (*n* = 45) metagenomic data were prepared using the BEST single-tube library preparation protocol[101] as optimized to be BGISEQ-500 compatible.[100] Briefly, genomic DNA was fragmented to 350 bp using an M220 Focused Ultrasonicator (Covaris, Woburn, MA). Sheared DNA was converted into BGISEQ-500 libraries following four steps: blunt end-repair, adapter ligation (2 $\mu$L of 10 × 10$^{-6}$ M BGI 2.0 adapters), fill-in reaction, and SPRI magnetic bead purification (Sigma-Aldrich). Indexing PCR cycle numbers for all metagenomic libraries (4–14 cycles) were determined through qPCR library quantification. Libraries were pooled equimolar over six lanes in 100bp or 150 bp paired-end mode on the BGISeq-500 platform aiming for a minimum of 50 million reads per sample.

*Assembly and Genome-Resolved Metagenomics*: Prior to sequence assembly, all paired-end reads were demultiplexed and quality filtered. AdapterRemoval v.2.3.1[102] was used to trim unidentified bases and adapter sequences from the ends of the reads. Host and human reads were removed using bwa-mem algorithm v.0.7.15[103] against the human (RefSeq: GCF_000001405.26) and red junglefowl (RefSeq: GCF_000002315.4) reference genomes. Quality filtered metagenomic reads were then coassembled using MEGAHIT v.1.1.1 with k-mer sizes: 77,87,97,107,127,137,147,157,167 and default parameters.[104] Contigs less than 2500 nt were removed from the resulting assembly output and corresponding header names were simplified using anvi'o v.6.2.[25] PCR duplicates were removed from the metagenomic reads used for coassembly with seqkit v.0.8.0[105] and subsequently mapped to the assembled contigs using bwa-mem algorithm v.0.7.15 with default parameters.[103] Samtools v.1.9[106] was used to sort and index the output SAM files into BAM files.

BAM files were used to generate a contigs depth of coverage table with jgi_summarize_bam_contig_depths (MetaBAT2 v.2.12.1).[107] We then applied the automatic binning algorithm in CONCOCT[108] on this coverage table to generate 10 large contig clusters to maximize explained patterns while minimizing fragmentation error, as performed elsewhere.[109,110] Subsequently, a manual binning and curation were performed for each CONCOCT cluster following the contigs workflow implemented in anvi'o v.6.2.[25] Briefly, anvi'o was used to generate a contigs database that identified open reading frames using Prodigal v.2.6.3[111] and single-copy core genes using HMMER v.3.2.1[112] against the collection of built-in HMM profiles for bacteria and archaea. Gene-level taxonomy was classified using Kaiju v.1.5.0,[113] with NCBI's nonredundant protein database, including fungi and microbial eukaryotes, and genes were further annotated with functions using the NCBI's Clusters of Orthologous Groups (COG).[114] Anvi'o was then used to profile each metagenomic BAM file to estimate the coverage and detection statistics of contigs in the contigs database and combined mapping profiles into a merged profile database for all individuals. In addition, one imported an anvi'o collection corresponding to the 10 CONCOCT clusters. Finally, each CONCOCT cluster was manually binned and further refined using the anvi'o interactive interface[115] taking into account sequence composition, differential coverage, GC content, and taxonomic signal of the considered contigs. MAGs with completeness >50% and redundancy <10% were retained for downstream analyses[116] (Genomic features of the MAGs can be found in Table S2b in the Supporting Information).

The taxonomy of the final list of MAGs was inferred using the Genome Taxonomy Database Toolkit (GTDB-Tk)[27] version 95. However, NCBI taxonomy was used from the GTDB output to describe the phylum of MAGs in the results and discussion sections, in order to be in line with the literature.

MAGs were considered to be detected in a given sample when >50% of their length was covered by reads to minimize nonspecific read recruitments.[110] The number of recruited reads below this cutoff was set to 0 before determining vertical coverage, the number of bases covering each genome divided by its length.

*Gut–Brain Module (GBM) Detection*: The red junglefowl shotgun metagenomic data were translated into neuroactive potential using a previously described module-based reconstruction framework.[24] Briefly, one searched for the presence of 56 gut–brain modules (GBMs), each corresponding to a process of synthesis or degradation of a neuroactive compound by the gut microbiota, in each of the red junglefowl MAGs (*n* = 192). As module structure follows the Kyoto Encyclopedia of Genes and Genomes (KEGG) database syntax, gene calls for each MAG were exported from the contig database within anvi'o and functionally annotated with KEGG identifiers using GhostKoala.[117] GBM coverage was calculated as the number of pathway steps for which at least one of the orthologous groups is found in a genome, divided by the total number of steps constituting the GBM using Omixer-RPM v.0.3.2 (https://github.com/raeslab/omixer-rpm). GBM presence in microbial MAGs was defined with a detection threshold of at least 66% coverage, to provide tolerance to miss annotations and missing data in incomplete genomes.[24] GBM detection was visualized with corrplot v.0.84[118] in the four differentially abundant or discriminant red junglefowl MAGs to identify over/underrepresented metabolic GBMs between the two behavioral selection lines.

*Differential Abundance Estimates*: Expected relative abundance of microbial taxa was modeled directly from read counts for 16S and shotgun sequence data at different taxonomic levels (phylum, class, order, family, genus, and ASVs) using a beta-binomial model controlling for collection year and controlling for the effect of selection and collection year on dispersion. The model was fit using corncob v.0.1.0,[119] an r-based package designed specifically for marker gene compositional data, which uses sophisticated models to account for sequencing depth and rare taxa in high-dimensional data and estimates abundance with uncertainties to support hypothesis testing between selection lines. The Wald test was used to test for differential taxon abundances between selection lines with a controlled false discovery rate (*p*-value cutoff < 0.05).[120] Graphical representations were performed in R using the package ggplot2 v.3.2.1.9000.[121]

## Supporting Information

Supporting Information is available from the Wiley Online Library or from the author.

## Acknowledgements

The authors thank Jacob Agerbo Rasmussen for his assistance in laboratory work and Antton Alberdi for his thoughtful guidance in data processing. Photo credit of red junglefowl image for the table of contents to P.J. The research was funded by The Danish National Research Foundation (grant no. DNRF143 to M.T.P.G.) and the Swedish Research Council (VR) (grant no 2015-05444 to P.J.).

## Conflict of Interest

The authors declare no conflict of interest.

## Data Availability Statement

The bacterial 16S rRNA gene and shotgun metagenomic sequence data that support the findings of this study are openly available in European Nucleotide Archive (ENA) at https://www.ebi.ac.uk/ena/, study accession number PRJEB46806.

## Peer Review

The peer review history for this article is available in the Supporting Information for this article.

## Keywords

behavioral selection, early domestication, fear, gut microbiota, metagenomics

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
