## [**Supplementary Information**: Record of Transparent Peer Review · Advanced Genetics]

Record of Transparent Peer Review

Gut microbiota linked with reduced fear of humans in Red Junglefowl has implications for early domestication

Lara C. Puetz*, Tom O. Delmont, Ostaizka Aizpurua, Chunxue Guo, Guojie Zhang, Rebecca Katajamaa, Per Jensen, M. Thomas P. Gilbert*

*Corresponding

Review timeline:

Date Submitted: 06 Aug 2021

Editorial Decision: 19 Aug 2021 Major revision

Revision Received: 17 Sep 2021

Editorial Decision: 22-Sep-2021 Accept pending formatting

Revision Received:

Accepted:

Editor: Myles Axton

Initial Editorial Evaluation

09 Aug 2021

Summary

Domestication of animals is enabled by selection for changes in wild behavior around humans. In some cases standing genomic variation may be selected or fixed, but in some cases, the rapidity of change in some domestication-related traits suggests selection for epigenetic modifications in behavioural or morphological gene expression. In this case, the contribution of gut microbiota was examined in a long running selection experiment in chicken domestication that resulted in two selected lines of red jungle fowl (wild ancestor of domestic chicken) that differ in their levels of fear in the presence of humans.

Scope

This is very much scope for Advanced Genetics. This evaluation across selection lines of the numerical abundance of chicken gut microbial taxa and their (gene expression) potential for synthesis of metabolites active in vertebrate (human and chicken) brain raises the possibility that selection can act upon gut microbiota or (not examined here) host gene variants influencing the relative abundance of certain taxa, notably Clostridiales and Lactobacilliales. It opens the door to GWAS of chicken lines with respect to microbiota as well as genetic and biochemical analysis of microbial metabolites on fear and fear extinction in domesticated animal species more generally.

1st Peer Review

10-18-Aug-2021

Reviewer #1

The gut-brain axis has been implicated in the host's behavior and cognition. In this study, Peutz et al. investigated the microbial differences between two lines of red junglefowl, which were selected for low or high fear of humans. The study revealed taxa differences between two lines. By using MGS data, the study further assessed the differences in the metabolic modules that may be implicated in the gut-brain axis. The study is well written and the analysis approaches were, in general, sound.

1.1 However, there is a big concern in the conclusion. In my opinion, the study cannot answer the question of whether the observed microbial difference is the outcome of the selection, the contributor of the behavior that is under selection, or a confounding effect of genetics.

1.1a Firstly, if the authors really want to answer how microbial changes under the behavioral selection, the authors should firstly also compare the gut microbiome at the P0 initial outbred parental generation. Does the microbial difference already exist between high or low fear junglefowl at the P0?

1.1b Secondly, during the subsequent breeding procedure, two lines of junglefowl may have different genetic backgrounds. It is also known that the host's genetics can influence the gut microbiome. It is also a question of whether the microbial difference is due to behavioral selection or genetic pressure.

1.1c Thirdly, it is not very clear to me how the red junglefowl was hatched and reared in mixed groups? Does it mean that two lines of junglefowl are mixed? Furthermore, if the microbial difference is indeed induced by the behavioral selection, how the

gut-brain modules are involved in this process? The discussed SCFAs are mainly produced by the gut microbiome, thereby cannot answer how behavioral selection affects the gut microbiome.

Specific comments:

1.2 The 16S analysis was done at the ASV level, while the discussion was done at the taxa level. The observation of ASVs cannot be directly generalized to the taxa level. For instance, 42 Firmicutes ASVs were significantly different. However, most detected ASVs were from Firmicutes too, as Fig 1 has shown and the authors have noticed. To conclude the effect of Firmicutes, the authors should combine all ASV from Firmicutes and do the analysis at the total Firmicute abundance. Therefore, it is unclear to me whether the presented data of Fig 2B & 2C were the abundance of Clostridiales and Lactobacilales ASV or the abundance of the taxa. The same for Fig3. If the authors present the data under the name of taxa, they should group all ASVs from the same taxa.

1.3 What were the read counts of 16S data? Did the authors rarify the read counts and ensure all samples have an equal number of reads? The sequencing depth has a big impact on the species abundance, especially on rare ones. The authors need to rarify reads to ensure the results of discriminant/rare ASVs convincing. Similarly, the proportion of host reads is a remarkable variable between samples, ranging from 19% to 91%. How did the read depth (after filtering the host reads) affect the detection of rare MGS?

Reviewer #2

This paper provides an assessment of microbiome changes associated with (relatively) short domestication selection experiment in Red Junglefowl. It utilizes samples from a unique experiment to conduct exploratory analyses into a potential role for microbiome changes in early animal domestication. The authors do an excellent job of presenting their results as exploratory and lay out briefly additional work that would be necessary to test for mechanism and causation. They also do a good job of presenting caveats in their discussion given the fact that the vast majority of the literature is on mammals not birds. Overall the writing, both content and tone, are well done.

I do have 3 significant suggestions and a handful of minor suggestions to improve the manuscript prior to publication.

2.1 There are currently no analyses of overall differences in microbial community or GBM composition and diversity. Both would be valuable for demonstrating whether the impacts observed are extensive or marginal and should be included whether or not they are significant. For instance, while there are some modules that significantly differ, it is unclear the value of having more of one or less of another if overall the vast majority are the same. Moreover, such analyses would help contextualize the work in the broader microbiome literature.

2.2 The first three figures are a bit of a muddle to me. It seems like Figure 1 is not the most informative/useful way to demonstrate who is differentially abundant especially since it doesn't distinguish between those more abundant in high vs low groups and there's no quantitative way to estimate what proportion are differentially abundant in panels with many many dots (like Firmicutes). Figures 2 and 3 do a better job of this but what they do which is different from one another is unclear. At the very least more information is needed in the figure legends (and potentially the results text) to explain what each figure is demonstrating.

2.3 The paper would benefit from a bit more discussion of how you expect domestication/selection on tameness actually acts to impact the microbiome. For instance, when you say "we hypothesized that host gut microbiota may be one of the earliest phenotypes to change as wild animals are domesticated" (line 59) do you mean that microbiota features are under selection or that they are responding to selection on host traits? How much would changes that accumulate over generations rely upon ongoing selection and/or vertical transmission fidelity and what about the domestication scenario (especially in fowl) promotes that? In your case, are both the high and low lines expected to diverge equally from an unselected wild state or is one (specifically the low fear which is more relevant to domesticated lineages) more impacted? While unselected or early generation samples are not sequenced here, discussion of how these data relate to published data on wild RJF would help answer at least some of these questions and others should be mentioned in the context of future work.

Minor suggestions:

2m1 Discuss the limitations of compositional data for inferring abundance differences. In an ideal world you'd have an estimate for absolute abundance (e.g. qPCR, sequencing spike in) but absent that, you should at least note the problems relative abundance data are known to have.

2m2 Note what test/method and threshold for significance was used to assess differential abundance in the results as well as in the methods.

2m3 Are more taxa differentially abundant in firmicutes than you would expect by chance given how many firmicutes there are overall? (lines 85-87)

2m4 Does "mixed-group pens" mean groups include both high and low fear individuals or just one or the other in groups? (line 65)

2m5 Specify N is number individuals they appear in or number of MAGs in that phylum? (line 135-138)

2m6 For the metagenomic data, did you test for an effect of library prep method (line 375-390)? Were all 6 samples prepped with the NEBNext protocol from one treatment group?

1 st Editorial Decision	19-Aug-2021
-------------

Editorial decision: Revise incorporating the reviewers' major comments and the editorial recommendations below

Editor's understanding of the reviews

Reviewer #1 Recommends Major Revision

Reviewer #2 Recommends Major Revision

These are the main reviewer recommendations that the editors believe will make the biggest improvement to this article. **Please do address all reviewer comments listed in the decision letter in your point-by-point response** (you may continue this table to do so if you wish). We hope this summary helps you to understand our decision and expedites the revision process. We value feedback from author and referees alike.
AdvGenet@wiley.com

Reviewer comments	Editor recommendation
1.1a Firstly, if the authors really want to answer how microbial changes under the behavioral selection, the authors should firstly also compare the gut microbiome at the P0 initial outbred parental generation. Does the microbial difference already exist between high or low fear junglefowl at the P0? 2.3 In your case, are both the high and low lines expected to diverge equally from an unselected wild state or is one (specifically the low fear which is more relevant to domesticated lineages) more impacted? While unselected or early generation samples are not sequenced here, discussion of how these data relate to published data on wild RJF would help answer at least some of these questions and others should be mentioned in the context of future work.	ED1 Use data from pre-selection animals or discuss published data to address the possible differences in microbiota of selected lines relative to unselected fowl. Are there differences in microbiota and fear responses between unselected animals? Do high and low selection lines differ in consistent ways from the preexisting microbe diversity?
1.1-In my opinion, the study cannot answer the question of whether the observed microbial difference is the outcome of the selection, the contributor of the behavior that is under selection, or a confounding effect of genetics. 1.1b- during the subsequent breeding procedure, two lines of junglefowl may have different genetic backgrounds. It is also known that the host's genetics can influence the gut microbiome. It is also a question of whether the microbial difference is due to behavioral selection or genetic pressure.	ED2 Would it be possible to compare the rates of change of the host genetic diversity in the two lines relative to the rate of change of microbial diversity?
1.2 The 16S analysis was done at the ASV level, while the discussion was done at the taxa level.	ED3 please carry out the analysis at the relevant taxonomic level since the reporting of these experiments is difficult to follow.

Author's Response to 1 st Review	17-Sep-2021
---	-------------

Reviewer comments	Editor recommendation	Author reply	Changes to Manuscript
-------------------	-----------------------	--------------	-----------------------

1.1a Firstly, if the authors really want to answer how microbial changes under the behavioral selection, the authors should firstly also compare the gut microbiome at the P0 initial outbred parental generation. Does the microbial difference already exist between high or low fear junglefowl at the P0? 2.3 In your case, are both the high and low lines expected to diverge equally from an unselected wild state or is one (specifically the low fear which is more relevant to domesticated lineages) more impacted? While unselected or early generation samples are not sequenced here, discussion of how these data relate to published data on wild RJF would help answer at least some of these questions and others should be mentioned in the context of future work.	ED1 Use data from pre-selection animals or discuss published data to address the possible differences in microbiota of selected lines relative to unselected fowl. Are there differences in microbiota and fear responses between unselected animals? Do high and low selection lines differ in consistent ways from the preexisting microbe diversity?	We agree that including gut microbiome data from the P0 generation or unselected fowls would yield a more thorough picture on the extent of how gut microbial community composition shifts during behavioral selection and early domestication. Unfortunately, at the time the experimental study began on the fowls, there was no plan to assess the gut microbial community and the animals are long since gone. One of the strengths of our data set is that the animals were reared in identical environmental conditions, including diet, and animals from both selections lines were housed together. Environment has previously been identified in the literature to cause large shifts in the gut microbial community, therefore including previously published data sets on wild red junglefowl into the statistical analysis would introduce large environmental biases and mask the fine scale differences we identified in our present study.	
1.1-In my opinion, the study cannot answer the question of whether the observed microbial difference is the outcome of the selection, the contributor of the behavior that is under selection, or a confounding effect of genetics. 1.1b- during the subsequent breeding procedure, two lines of junglefowl may have different genetic backgrounds. It is also known that the host's genetics can influence the gut microbiome. It is also a question of whether the microbial difference is due	ED2 Would it be possible to compare the rates of change of the host genetic diversity in the two lines relative to the rate of change of microbial diversity?	Unfortunately this is not feasible at this time as the time series data on both the microbial and host genetic diversity do not exist.	

to behavioral selection or genetic pressure.			
1.2 The 16S analysis was done at the ASV level, while the discussion was done at the taxa level.	ED3 please carry out the analysis at the relevant taxonomic level since the reporting of these experiments is difficult to follow.	We did run statistical analyses at the different taxonomic levels and not just at the level of the ASVs. We apologize for not clearly stating this in the original text.	The text was re-written to clarify models and test statistics used for hypothesis testing of differential abundance for 16S and shotgun data in the methods section. We report test statistics and clarify the taxonomic level corresponding to each analysis presented in the results and figure legends throughout the manuscript.

Reviewer #1

The gut-brain axis has been implicated in the host's behavior and cognition. In this study, Peutz et al. investigated the microbial differences between two lines of red junglefowl, which were selected for low or high fear of humans. The study revealed taxa differences between two lines. By using MGS data, the study further assessed the differences in the metabolic modules that may be implicated in the gut-brain axis. The study is well written and the analysis approaches were, in general, sound.

1.1 However, there is a big concern in the conclusion. In my opinion, the study cannot answer the question of whether the observed microbial difference is the outcome of the selection, the contributor of the behavior that is under selection, or a confounding effect of genetics.

1.1a Firstly, if the authors really want to answer how microbial changes under the behavioral selection, the authors should firstly also compare the gut microbiome at the P0 initial outbred parental generation. Does the microbial difference already exist between high or low fear junglefowl at the P0?

- We agree with the reviewer that including data from the P0 generation or data on unselected fowls would yield a more thorough picture on the extent of how gut microbial community composition shifts during behavioral selection and early domestication. Unfortunately, at the time the experimental study on the fowls began, there was no plan to assess the gut microbial community and the animals are long since gone. We still find it interesting that differences in the gut microbiota exist between the two selection lines and are already identifiable at such an early stage in the domestication process.

1.1b Secondly, during the subsequent breeding procedure, two lines of junglefowl may have different genetic backgrounds. It is also known that the host's genetics can influence the gut microbiome. It is also a question of whether the microbial difference is due to behavioral selection or genetic pressure.

- We agree with the reviewer's comment that it would be really interesting to assess how the host genetic diversity between the two lines changes relative the rate of change of the microbial diversity. Unfortunately this is not feasible at this time as the time series data on both the microbial and host genetic diversity do not exist. We try to present our results as exploratory based on the limitation of our data set in the manuscript and briefly discuss additional work that would be necessary to test for mechanism and causation in the discussion.
- Incidentally, for the reviewer's interest, we can inform them that some very preliminary data analysis done by colleagues on the sequences suggest that the fowl genomes do not separate based on selection line. However, given this is beyond the scope of this study and is hardly a robust observation, we have elected not to include mention of it here.

1.1c Thirdly, it is not very clear to me how the red junglefowl was hatched and reared in mixed groups? Does it mean that two lines of junglefowl are mixed? Furthermore, if the microbial difference is indeed induced by the behavioral selection, how the gut-brain modules are involved in this process? The discussed SCFAs are mainly produced by the gut microbiome, thereby cannot answer how behavioral selection affects the gut microbiome.

- We apologize for the lack of clarity. “Mixed-group pens” means that both high and low fear fowls for a given generation were housed together throughout the study from when hatched onwards throughout their life. We have amended the text to clearly describe the meaning of “mixed-group pens” in the methods section (see lines 313-316). Also, if the reviewer is interested, full details on the breeding scheme, selection process and housing can be found in previously published work describing the onset of the experiment (see lines 306-308 for appropriate references). [Refs 21-23 in MS18.R1-ED]
- In this exploratory study, one of the main goals was to identify if differences in the microbial community composition exist between the two selection lines and if yes, what were some of the implications that these differences may have in regards to behavioral selection and early domestication of red junglefowl. We know from previously published studies that gut bacteria are strongly implicated in brain development, host behaviour, and cognition, therefore, it is possible that when selecting specifically for behavior in the host, we are unintentionally selecting for certain host associated bacteria that contribute to fear related behavior. Exact mechanisms through which the gut microbiome contribute to the gut-brain-axis, and hence behavior, remain elusive. However, extensive research has associated certain gut bacteria, including neuroactive compounds that are produced in the gut by bacteria, with host behaviors including fear. In this regard, including gut-brain modules (e.g. SCFAs) in the discussion of the manuscript is relevant to understanding their potential roles in animal behavior associated with domesticates and provide further insight into the mechanistic explanations of the domestication process overall.

Specific comments:

1.2 The 16S analysis was done at the ASV level, while the discussion was done at the taxa level. The observation of ASVs cannot be directly generalized to the taxa level. For instance, 42 Firmicutes ASVs were significantly different. However, most detected ASVs were from Firmicutes too, as Fig 1 has shown and the authors have noticed. To conclude the effect of Firmicutes, the authors should combine all ASV from Firmicutes and do the analysis at the total Firmicute abundance. Therefore, it is unclear to me whether the presented data of Fig 2B & 2C were the abundance of Clostridiales and Lactobacilales ASV or the abundance of the taxa. The same for Fig3. If the authors present the data under the name of taxa, they should group all ASVs from the same taxa.

- We thank the reviewer for bringing to our attention the need for further details on the statistical analyses that were performed on the 16S amplicon sequence data. Differential abundance was indeed performed at each of the different taxonomical levels that were presented in the results section of the original manuscript. We amended the text and figure legends to clarify at which taxonomic level each differential abundance analysis was performed at included more details in the methods section (see lines 424-433).

1.3 What were the read counts of 16S data? Did the authors rarefy the read counts and ensure all samples have an equal number of reads? The sequencing depth has a big impact on the species abundance, especially on rare ones. The authors need to rarefy reads to ensure the results of discriminant/rare ASVs convincing. Similarly, the proportion of host reads is a remarkable variable between samples, ranging from 19% to 91%. How did the read depth (after filtering the host reads) affect the detection of rare MGS?

- While we appreciate that there are different ways to normalize 16S and shotgun sequence data, we disagree with the reviewer and chose not to rarefy the read counts due to the biases that this type of normalization can introduce to the data (see McCurdie and Holmes, 2014; Willis 2019a). We instead opted to model abundance with a beta-binomial regression, which models parameters based on errors estimation and compares diversity estimates relative to these errors. These types of models are incorporated into the recently published corncob package that was used for the statistical analyses presented in the manuscript and can be performed directly on read counts for both 16S and shotgun data (see in lines 424-430). Using this approach, we were able to adjust for sample size, sequencing depth and unobserved taxa when comparing microbial abundances between the two different selection lines without discarding data (Martin *et al.*, 2020). [Reference 119 – ED]

<https://doi.org/10.1371/journal.pcbi.1003531>

<https://doi.org/10.3389/fmicb.2019.02407>

<https://doi.org/10.1214/19-AOAS1283> [Reference 119 – ED]

- A minimum number of reads after all bioinformatics filtering steps was required for a given sample to be included in the downstream statistical analyses. For 16S sequence data samples with <10,000 reads were removed from the analysis (see line 350-351, and 10 million single-end reads (1 Gb) was required for the shotgun sequence data (see lines 118-120), both of which are standard in the literature.
- Read counts per sample are provided in the Supplementary tables 1 & 2 for 16S and shotgun data respectively.

Reviewer #2

This paper provides an assessment of microbiome changes associated with (relatively) short domestication selection experiment in Red Junglefowl. It utilizes samples from a unique experiment to conduct exploratory analyses into a potential role for microbiome changes in early animal domestication. The authors do an excellent job of presenting their results as exploratory and lay out briefly additional work that would be necessary to test for mechanism and causation. They also do a good job of presenting caveats in their discussion given the fact that the vast majority of the literature is on mammals not birds. Overall, the writing, both content and tone, are well done.

- We thank the reviewer for their kind words of support and nice summary of the work.

I do have 3 significant suggestions and a handful of minor suggestions to improve the manuscript prior to publication.

2.1 There are currently no analyses of overall differences in microbial community or GBM composition and diversity. Both would be valuable for demonstrating whether the impacts observed are extensive of marginal and should be included whether nor they are significant. For instance, while there are some modules that significantly differ, it is unclear the value of having more of one or less of another if overall the vast majority are the same. Moreover, such analyses would help contextualize the work in the broader microbiome literature.

- In addition to figure Fig.4 (now Fig.3 in the revised manuscript) describing the occurrence of GBMs by category across all MAGs, we have now added an additional presence/absence table of GBMs per MAG for each sample (see Suppl. Table 2C). Furthermore, we include an additional figure in the manuscript (Fig. 4C) describing the frequency of each GBM per sample and highlight the selection line and generation within the figure.
- Regarding the microbial community composition, in addition to Fig 1A describing overall relative abundance of microbial community composition at the order level, we have added ordination plots in the supplementary material for both 16S and shotgun data (Suppl. Fig. 2). There is no clear shift between the two cohorts at this level of diversity.

2.2 The first three figures are a bit of a muddle to me. It seems like Figure 1 is not the most informative/useful way to demonstrate who is differentially abundant especially since it doesn't distinguish between those more abundant in high vs low groups and there's no quantitative way to estimate what proportion are differentially abundant in panels with many many dots (like Firmicutes). Figures 2 and 3 do a better job of this but what they do which is different from one another is unclear. At the very least more information is needed in the figure legends (and potentially the results text) to explain what each figure is demonstrating.

- We apologize for the confusion. We agree with the reviewer's comments in regards to the significance of Fig 1 in the main manuscript and have now moved into the supplementary material section. The figure was meant to provide an overview of the overall community composition in red junglefowl at the phylum level and highlight that differentially occurring ASVs were among highly prevalent as well as rare members of the gut bacterial community between selection lines.
- The remaining two figures now included more details in the figure legends in order to clearly distinguish the difference between the results. Fig 2 (now Fig. 1 in the revised manuscript) represents the differential abundance of ASVs agglomerated at the order level and at the class level for the MAGs that were statistically significant between the two selection lines. Fig 3 (now Fig.2 in the revised manuscript) represents differentially abundant or discriminant ASVs associated with either selection line that were statistically significant.

2.3 The paper would benefit from a bit more discussion of how you expect domestication/selection on tameness actually acts to impact the microbiome. For instance, when you say "we hypothesized that host gut microbiota may be one of the earliest phenotypes to change as wild animals are domesticated" (line 59) do you mean that microbiota features are under selection **or that they are responding to selection on host traits?**

- This is a very good question and unfortunately we cannot identify which scenario is the case based on the data set we present in the manuscript. Please see the second response to the reviewer comment (1.1c) above. We thank the reviewer for the suggestion and have altered text in the discussion to clarify (please see lines 332-334).

How much would changes that accumulate over generations rely upon ongoing selection and/or vertical transmission fidelity and what about the domestication scenario (especially in fowl) promotes that? In your case, are both the high and low lines

expected to diverge equally from an unselected wild state or is one (specifically the low fear which is more relevant to domesticated lineages) more impacted? While unselected or early generation samples are not sequenced here, discussion of how these data relate to published data on wild RJF would help answer at least some of these questions and others should be mentioned in the context of future work.

- These are really great questions. In theory, we do believe that the two selection lines are supposed to diverge equally, although in reality, that is of course and empirical question. Although we do not have the gut microbiome data to support this, we see that behaviorally, physiologically and morphologically, the two selection lines of red junglefowl show relatively equal deviation from the unselected population in earlier generations (references provided below). In both of these studies, the unselected line of red junglefowl resembled the parental line in most traits whereas the two selected lines diverged equally from that.

<https://doi.org/10.1371/journal.pone.0035162> [Reference 22 in revised MS - ED]

<https://doi.org/10.1038/s41598-017-03236-4> [Reference 53 in revised MS- ED]

- Unfortunately, the unselected line ended by generation 6, due to a lack of resources, and thus we cannot do the comparisons today.
- One of the strengths of our data set is that the animals were reared in identical environmental conditions, including diet, and animals from both selections lines were housed together. Environment has previously been identified in the literature to cause large shifts in the gut microbial community therefore including previously published data sets on wild red junglefowl into our current statistical analysis would introduce large environmental biases and mask the fine scale differences we identified in our present study.

Minor suggestions:

2m1 Discuss the limitations of compositional data for inferring abundance differences. In an ideal world you'd have an estimate for absolute abundance (e.g. qPCR, sequencing spike in) but absent that, you should at least note the problems relative abundance data are known to have.

- We thank the reviewer for the suggestion and have expanded on the limitations these types of data sets include in the differential abundance subsection of the methods. (See lines 427-431)

2m2 Note what test/method and threshold for significance was used to assess differential abundance in the results as well as in the methods.

- We have now included the test statistic and significant threshold cutoffs in the results and figure legends and provide more detail in the methods section of the manuscript (see lines 424-432).

2m3 Are more taxa differentially abundant in firmicutes than you would expect by chance given how many firmicutes there are overall? (lines 85-87)

- We agree that more differences are likely to be detected in the statistical analyses the more prevalent a taxa is. None the less, we still identified many significant differences in ASVs from less abundant phylum. Firmicutes were not themselves differentially abundant when agglomerating all ASVs at the phylum level. However, many of the significant differences in the differential abundance analyses (at the order, genus and ASV level) belonged to Firmicutes. We have altered the text in the results section to describe differences at the lower taxonomic levels instead.

2m4 Does "mixed-group pens" mean groups include both high and low fear individuals or just one or the other in groups? (line 65)

- "Mixed-group pens" does indeed mean that both high and low fear fowls for a given generation were housed together throughout the study from when hatched onwards throughout their life. We have amended the text to clearly describe the meaning of "mixed-group pens" in the methods section. (see lines 313-316)

2m5 Specify N is number individuals they appear in or number of MAGs in that phylum? (line 135-138)

- Thank you for the suggestion. We have clarified in text. (line 126)

2m6 For the metagenomic data, did you test for an effect of library prep method (line 375-390)? Were all 6 samples prepped with the NEBNext protocol from one treatment group?

- We did test for the effects of the different library preparation methods on the shotgun sequence data. Please see attached figures below that were not included in the manuscript. The shotgun library preparation method did not appear to bias the percent of reads per sample that mapped to the assembly nor the collection of MAGs (Fig. A&B). Furthermore, the principle components analysis of the log-ratios of the number of reads mapped to each of the MAGs per sample did not show clear separation of samples prepared using the NebNext protocol from those using the BEST-BGI library preparation protocol (Fig. C).
- The 6 samples prepared with the NebNext protocol were initially chosen for the pilot study and represented an equal number of individuals from each selection line (HF: n=3; LF: n=3)(Fig.B).

2nd Editorial Decision

22-Sep-2021

The manuscript has now been extensively revised incorporating all the comments of the two reviewers. The editor believes the authors have addressed the reviewer comments to clarify their article and have done what they can within the scope of their experimental setup. We have now decided to accept the revised manuscript in principle, subject to the attached formatting requirements and data access provisions.

Accept for publication

[date]